# The Role of 'Influencers' as Drivers of a More Sustainable Urban Freight Sector

**Alena Brettmo and Jon Williamsson \***

School of Business, Economics and Law, Department of Business Administration, Division of Industrial and Financial Management and Logistics, University of Gothenburg, P.O. Box 610, SE-40530 Gothenburg, Sweden; alena.brettmo@handels.gu.se
\* Correspondence: jon.williamsson@handels.gu.se

**Abstract:** The importance of stakeholders in the development of a sustainable urban freight sector has been highlighted in recent research. Not all stakeholders have a direct link to the supply chain, but they may still play a role in creating sustainable urban freight initiatives. This study explores the initiatives that norm-setting indirect stakeholders in urban freight, referred to as 'influencers', establish to support a more sustainable urban freight sector, and how those initiatives may impact the business models of carriers. The study uses data gathered for ongoing research into the roles of indirect stakeholders in the development of sustainable urban freight initiatives. The results indicate that influencers can put pressure on receivers regarding logistical issues and shape the physical environment in which deliveries are conducted. Influencers use three primary strategies to support sustainable urban freight: vehicle-focused measures, consolidation linked to physical infrastructure, and consolidation through behavioural changes. These initiatives impact the relationship between receivers and carriers and may push carriers to adopt more sustainable practices as well as take decisions that impact their business models. The results highlight the often-overlooked power of influencers in relation to the development of actor behaviour in the urban freight supply chain and show the potential for both conflict and change arising from the use of this power.

**Keywords:** urban freight; stakeholder; influencer; business model

## 1. Introduction

Stakeholders that are not directly involved in the supply chain but whose actions have a bearing on several characteristics of the local transport system have recently been highlighted in urban freight research, e.g. Reference [1]. Business improvement districts (BIDs), facility management companies and public procurement organisations are examples of stakeholder groups that at first glance appear peripheral but who actually may exert considerable influence on more central figures in the urban freight sector, cf. References [2,3]. In relation to key players (actors) in the supply chain, these remote stakeholders (The definitions of stakeholder and actor in this article were adopted from Ballantyne et al. (2013). Actors are organisations that have direct influence on the urban freight system; stakeholders are organisations that have interests (direct and indirect) in urban freight.) are indirect in the sense that they are not directly responsible for the sale, transportation, or ordering of products. These organisations influence other actors such as goods receivers on how they organise their logistical flows and procurement practices. This is achieved by, for example, providing receivers with guidelines for the selection of service carriers or by setting rules for delivery practices within specific areas [4]. Other concrete examples of stakeholders are property owners that design and control access to freight-related infrastructure or public procurement associations that put restrictions on the type of vehicles eligible for certain delivery assignments and stipulate rules for regulating the time of deliveries [5]. Hence, such

third-party organisations both influence the types of services available within certain areas and shape the physical aspects of the urban milieu, such as the location and size of loading facilities. Considering the power of these norm-setting indirect stakeholders [1] in relation to the actors more central to the supply chain, we suggest that they could be labelled as influencers. These organisations tend to engage with issues linked to sustainability (e.g., local pollution or congestion) and therefore are often strongly motivated to enforce standards of behaviour that by changing the attitudes of receivers may put financial pressure on the firms providing transport services (i.e., carriers) [5]. However, the role of these organisations in relation to the business side of transport services is still poorly understood.

Urban freight research has recently emphasised the importance of taking business practices into consideration when moving sustainable innovations from a conceptual stage to the market; to execute this research, the business model, which is a comparatively new analytical concept, has been adopted. Research that relies on the business model has been motivated by an ambition to develop a holistic understanding of what inspires businesses to deploy new types of services, e.g., Reference [6], or adopt technological innovations, e.g., Reference [7]. By highlighting the business model, research shifts focus from single-factor explanations, such as pricing, to more complex relationships related to the topics of corporate value creation, stakeholder relationships, and corporate sustainability [8,9]. This study aims to explore the roles of influencers in promoting sustainable urban freight activities and to identify the potential impact of their initiatives on the business models of the carriers that are active in an urban freight setting. Fulfilling the purpose, the study paves the way for research that evaluates the impact of influencers' sustainability measures on carriers, as well as investigation of how influencers may support the development of carriers' business models in ways that will facilitate the creation of a more sustainable urban freight sector.

## 2. Literature on Business Models, Stakeholders and Influencers in Urban Freight

The introduction of the business model concept in research on urban freight seems to be largely based on the expectation that an innovation may move past the demonstration stage and achieve long-term change only when the business sector has access to a viable business model. This perspective is in line with contemporary business model research and is based on the functionality that has been attributed to the concept in previous research [10]. In urban freight research, the business model concept has been accentuated as a necessary factor in the introduction of technological innovations, such as electric vehicles [11], and it has been argued to play a key role when trying to bring about changes in how companies arrange their operations [7]. The search for a viable business model has been identified as a significant challenge for companies providing new types of logistics services, such as instant deliveries [12,13]. In addition, research indicates that business models have a potential role to fulfil in relation to the realization of sustainability initiatives in cities, such as joint delivery systems [14] and consolidation services [6,15].

The business model concept is generally considered as a means to define both the organisational and financial architecture that a business rests upon, and the business model is deployed in a way that demonstrates how the business creates and delivers value to customers [8]. As such, the business model concept becomes a broad description of what makes a business function in a specific setting or with certain types of resources or processes in place [10]. A generally accepted definition and widely used application of the business model concept is the business model canvas (BMC) [16] that has been applied in earlier urban freight research; cf. Reference [7]. The BMC framework provides a detailed understanding of a business model by dividing it into blocks that are mapped using pre-specified questions. These questions are linked to both practical and theoretical aspects of business management [16], implying that the BMC builds on an eclectic selection of perspectives on what it is that drives a company and enables it to become profitable. The BMC comprises nine building blocks grouped into four sections that describe the key aspects of business models [16]: (i) customers (a description of customers segments, the customer relationships and channels used to interact with the customers), (ii) offering (a description of the value offering that the company presents to its customers),

(iii) infrastructure (a description of the key activities, key resources and key partners needed to produce the offering) and (iv) finances (describes the revenue streams and the cost structure associated with the business model).

These four sections show that the BMC includes descriptions of actors that are important to the business: for example, the customer and suppliers. The BMC thus explicitly states what makes a business model attractive to different actors and a wide range of stakeholders. As such, describing the qualitative dimensions linked to value creation is necessary. This makes business models difficult to evaluate based on easily-accessible quantitative data, such as accounting and production data [8]. Instead, to understand a company's business model it is necessary to verbalise, visualise, and understand different stakeholders' incentives to interact with the company. Accordingly, the business model can be said to describe a company's raison d'être [17].

*Research on Influencing Organisations as a Group of Stakeholders in Urban Freight*

Research on stakeholders in urban freight has been growing over the past 10–15 years [1,18–20]. Researchers have acknowledged the importance of considering the opinions and rationales of each stakeholder group, even though stakeholders' rationales may contradict each other [20]. A number of proposals have suggested how to best take the opinions of stakeholders into account during the planning process or how to include stakeholders in a broader governance perspective via tools that generate the optimal solution or support the development of a consensus between stakeholders [21–23]. It has been proposed that the application of discrete choice and agent-based modelling could support decision-makers during the policy design process by providing them with means to evaluate the implications of changes to each stakeholder group [24,25]. Emphasis has, however, been on the importance of building relationships through collaboration and partnerships between stakeholder groups [1,19,20,26]. Approaches vary in terms of which participants in the urban freight setting should be considered as stakeholders, and research acknowledges the existence of heterogeneity in the characteristics, interests, and behaviours of stakeholders [1,27,28]. It has thus been proposed that the group of relevant stakeholders that should be considered in urban freight should be expanded by including groups with indirect interests in urban freight, such as property owners, commercial organisations and trade associations [1].

As described in the introduction, we define influencers as norm-setters that are third-party in relationship with regard to the actors directly involved in the urban supply chain, cf. Reference [1]. Influencers are thus a group of stakeholders who often may not directly be involved in the sending or receiving of goods, nor do they take the title of the goods. Yet, their actions affect how supply chain actors manage their goods flows. In Figure 1, we conceptualise influencers as organisations that exert direct influence on how especially small goods receivers may operate and arrange their logistics related activities and indirect influence on other actors in the supply chain. Previous research indicates that influencers engage with issues linked to sustainability (e.g., local pollution or congestion) and enforce standards of behaviour on the receiver [5].

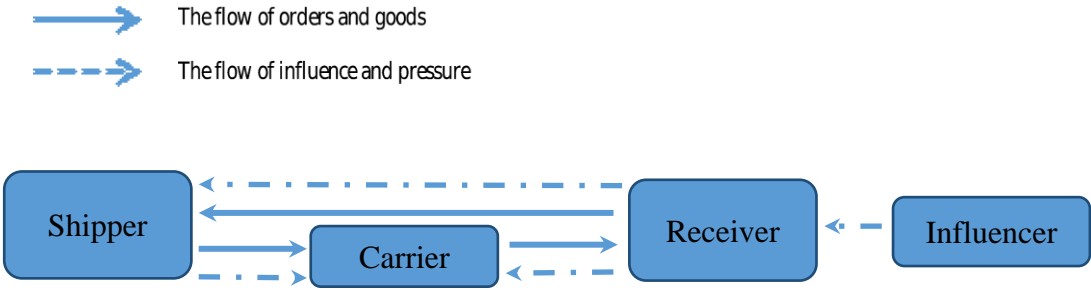

**Figure 1.** The influencers' role in relation to the supply chain.

In Figure 1, the solid lines represent the flow of orders and goods and dotted lines present the flow of pressure or influence. Note that in Figure 1, the pressure is shown as being unidirectional; however, the interaction between receivers and influencers is likely built on negotiation and involves the demands of many types of stakeholders. If the receiver´s logistical preferences change, a chain of events is expected to occur in the supply chain. First, contracts between the receiver and shipper must be amended. Then, the shipper must put pressure on the carrier to make changes in how the transport service is executed. However, note that the receiver also may directly put pressure on the carrier despite not having a direct contractual relationship. Consequently, the development of pressure is a complex and poorly understood process, and this research will help in formalising the boundaries within which potential influence is to be expected.

In the literature, four key groups of influencers have been identified as active in the urban freight setting [29]: BIDs and similar initiatives, property owners, facility management companies (FMCs), and public procurement organisations (PPOs).

BIDs are partnerships between public and private sector actors established to improve the attractiveness of a specific geographic area in ways that benefit the local business sector [30,31]. BIDs have gradually grown in popularity and become an international phenomenon that has been adapted to local requirements [32]. Some actors within BIDs are aware of the externalities caused by extensive urban freight traffic and strive to target those externalities to make the local area more attractive for customers and visitors [29]. BIDs in different countries are engaged in a wide range of activities; some of these activities have a significant impact on urban freight. The range of activities that BIDs are engaging in and the level of involvement in urban freight issues depends on the context and the main stakeholders that they represent, factors which usually vary from country to country. At the same time, BIDs in differ countries share many characteristics and contextual factors, which means that they still can be seen as one type of organisational category [29]. BIDs unite fragmented actors, especially businesses, and promote sustainable urban freight solutions by, for example, providing platforms to implement pilot projects or schemes that lead to more sustainable deliveries in urban areas [2]. Tangible initiatives include freight consolidating schemes, the introduction of environmentally-friendly vehicles for the final deliveries, the establishment of small-scale urban consolidation centres (UCCs) that decrease the amount of traffic in the area, collaborative procurement schemes such as buyers' clubs that reduce the number of suppliers involved and result in fewer trips, common recycling and waste management schemes, discouraging of private deliveries to workplaces, and the development and promotion of delivery and servicing plans (DSP) [2,5,33].

Property owners are considered as a powerful and important group of influencers with incentives which are active regarding issues related to urban freight. The externalities generated by urban freight may have a detrimental effect on property value and thus impact the balance sheet of property owners. Property owners have the power to promote or even dictate to their tenants how to organise their logistics setup or the parts of it connected to flows in and out of buildings. Examples of tenants that make logistical decisions are offices, restaurants, and shops. By promoting and facilitating the consolidation of flows, organising an accessible goods reception area, and extending concierge services for goods, property owners may facilitate the changes in delivery routines that make them more sustainable. Property owners may, for example, persuade tenants in a multitenant building to use the same facility management company; this would lead to the consolidation of both flows going in and out of the building. In Sweden, the project Älskade stad showed that property owners in Stockholm can take the lead in the process of implementing sustainable solutions for their building management and services provided to tenants [34]. Consequently, property owners can be aware of and act on the negative impacts of externalities generated by urban goods movements. They also appear to understand that accessibility for the delivery of goods (i.e., goods are efficiently delivered without jeopardising air quality, safety and liveability) plays a crucial role for a well-functioning city by enhancing the quality of the area in which their property is located. Such awareness is tightly linked to the interest of property owners to secure the future commercial value of their properties.

FMCs work to ensure certain standards in facility management by providing services and products to tenants. FMCs often manage certain logistical flows to and from their customers, provide cleaning and catering services, and at times even take over purchasing functions for some goods. FMCs thus provide a wide range of services for their customers, and some FMCs establish a green profile by catering to certain aspects of sustainability when interacting with customers. Previous research has shown that FMCs prioritise certain environmental aspects of their operations and are striving for continuous improvement in their environmental performance [4]. They often use environmental improvements as a marketing strategy. This allows them to position themselves in relation to competitors by adding value to the service portfolio, thus giving them a competitive edge. FMCs may act as orchestrators of logistics flows on behalf of their customers by choosing the most sustainable suppliers and optimising goods movements [4]. In particular, by managing several customers in the same multi-tenant building or managing several buildings in a small geographical area, this category of stakeholders may have a considerable impact on large volumes of goods. However, although FMCs are willing to work for achieving sustainability, they often need the support of the property owners. This is especially the case when investment is needed in the property to decrease the externalities that freight movements create.

PPOs are organisations that have an 'umbrella' function, providing a frame of reference for the goods that will be delivered to public organisations, such as schools or nursing homes. PPOs decide on the type and quality of products and services that should be provided to the final customers; they organise tenders, choose the suppliers and create framework agreements that stipulate how the deliveries should be made. Their policy regarding the deliveries of goods can make a significant impact, leading to an improvement in urban deliveries by making them more sustainable [4]. PPOs are cost-conscious but often motivated by social and environmental values; that is, by doing the right things in the eyes of their clients. This means that they often prioritise quality and sustainability [4]. Several studies have described the connection between purchasing practices and freight trip generation. One of the key findings of these studies is that public spending is usually important and thus can leverage the application of environmentally-sustainable deliveries of goods for public purposes; for example, using more green vehicles or sustainable delivery schemes [35,36]. A case study on the procurement practices of municipal PPOs showed that their decisions on choosing and contracting supplier and goods specifications impact the freight flows for public needs by specifying the delivery terms, the environmental standards of the vehicles, and the frequency of deliveries made, as well as by including incitement for demand planning and demand consolidation, optimising delivery schedules and routes, and coordinating deliveries when possible [4].

## 3. Materials and Methods

This paper reports on results from an ongoing study about indirect stakeholders in urban freight started in 2015. The initial research focused on goods receivers [4,5] and showed that other organisations influence the patterns of urban freight movements and how deliveries are made in urban centres. A series of interviews was conducted with different types of influencing organisations identified using the actors–resources–activities (ARA) methodology from the industrial network approach [37]. ARA supports the collection of data on actors and the links between them as well as the resources and activities associated with the networks that arise from the interactions between the actors. As such, ARA shares similarities with the BMC in the sense that both tools aim to portray incentives for interaction. Therefore, the data collected through ARA can be analysed using a BMC because the principal themes of the business model (i.e., the logic behind activities and interactions) were included in the data set developed through the ARA approach.

The interviews were conducted between 2015 and 2018. In total, representatives from 21 organisations were interviewed (See Table 1). Out of those 15 were classified as influencers, five had direct interests in urban freight (i.e., this category contains two municipal traffic agencies and three carriers), and one was an umbrella organisation for BIDs.

**Table 1.** The list of organisations interviewed for the study.

| Influencers | Country |
| --- | --- |
| Four Property owners/property developers | Sweden |
| Two FMCs (one organisation with two sites) | Sweden |
| Two PPOs | Sweden |
| Other actors or stakeholders | |
| One umbrella organisation for BID-like associations | Sweden |
| Three carriers, including two cargo bike delivery companies | Sweden |
| Two municipal traffic agencies | Sweden |

Representatives from three BIDs in the UK and four BIDs in the US were interviewed. The BIDs were chosen based on their past or current involvement in urban freight initiatives or interest to participate in future initiatives; therefore, the idea was to consider only the BIDs interested in urban freight. In Sweden, interviews were conducted with the representatives from one BID-like association, an FMC with two different sites, four property owners/developers, one umbrella organisation for BIDs, two PPOs, and two municipal traffic agencies. In addition, interviews were conducted with a carrier and two cargo bike delivery companies to obtain their views on the reality of providing services in areas where influencers are active. These interviews were also used to understand the business models of carriers active in urban freight. Interviews were conducted with senior and middle-level managers, and at several instances, interviews were conducted with multiple respondents from the same organisation. The FMC has customers in all four Nordic countries. The activities of PPO depend a lot on country legislation, but in general, their work is regulated by EU legislation on public procurement (2004/18/EC directive). However, the distribution of interviews between the countries implies that the Swedish and Scandinavian perspective dominates when it comes to property owners, FMCs and PPOs, while the results from the BIDs are coloured by American and U.K. perspectives.

To analyse the potential impact on how carriers run their businesses, the empirical material was scrutinised using the BMC as the principal tool. The data collection and analysis were conducted using the following steps, which are also described in Figure 2.

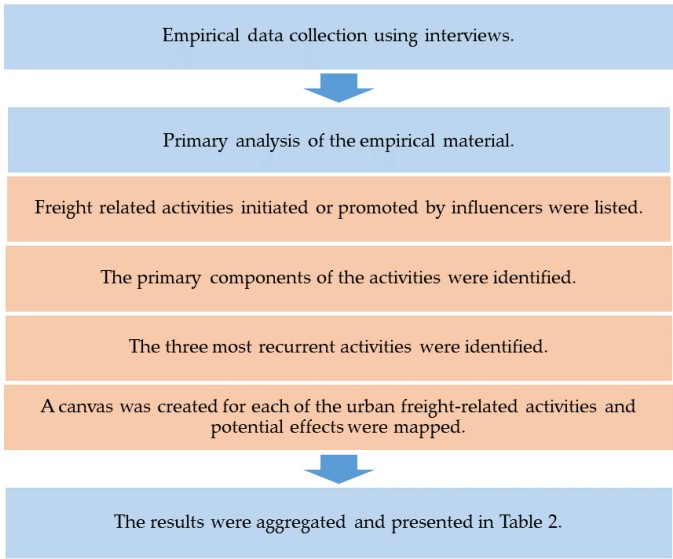

**Figure 2.** The empirical research process.

First, we listed the freight-related activities initiated or promoted by influencers by matching the empirical material against previous research [2]. To limit the scope of the study we then identified the primary components of the activities, as well as the three most recurrent principal activities. Next, we explored how the activities potentially impact carriers´ business models by mapping the potential for influence on the BMC of a typical carrier. We plotted the potential effects of each principal activity on

each block of the BMC and then summarized the findings in Table 2. In our BMC canvas, we merged customer relations and channels because these two blocks are closely interconnected and serve a similar purpose, which is to clarify how the interaction between the carrier and customer functions [16]. The approach turns the traditional urban freight perspective upside-down by using remote stakeholders as the starting point. However, using this approach, it is difficult to look beyond the activities and conditions within the scope of the influencers. Hence, the focus was on identifying how the initiatives of influencers may, directly and indirectly, impact activities and conditions, as seen in Figure 1.

## 4. Results

At an aggregated level, the matching of the identified initiatives against those found in previous research [2] produced the following list of types of initiatives that the influencers engaged with:

- Vehicle-related requirements such as greener vehicles, e.g., low emissions vehicles, hybrid and electrical vehicles
- Requirements to participate in (e.g., deliver to) UCC initiatives
- Consolidating deliveries and collaborative procurement
- Scheduling or time-agreed deliveries, off-hour deliveries (OHD)
- Common carrier locker system
- Deliveries to pick-up point
- Parking-related regulations and anti-idling programs
- Delivery and servicing planning (DSP)

Next, we show how influencers have the potential to impact goods receivers and the setting in which deliveries are made in ways that influence the carriers providing delivery services to receivers. For each category of influencers, we describe the main activities (urban freight initiatives) promoted and supported by the influencing organisations. Selected quotes from the interviews can be found in Appendix A.

### 4.1. Business Improvement Districts

The focus was on the BIDs that have a certain interest and/or involvement in urban freight initiatives. The main urban freight initiatives, herein called activities, linked to these BIDs were as follows:

(1) Collaborative procurement via so-called buyers' clubs. This is a service provided to BID members that gives them the possibility to procure some common types of goods and services with discounted prices from preferred suppliers. Reduced numbers of suppliers can lead to a reduced amount of delivery trips and deliveries to several receivers at once.

(2) Waste consolidation and associated delivery vehicle reduction schemes. This is a two-phase program that focuses on waste vehicle reduction during the first phase and adds on the reduction of commercial vehicles during the second phase via the introduction of consolidation schemes.

(3) Freight consolidation schemes using UCC and delivery to the final goods receivers using environmentally-friendly vehicles.

(4) Facilitation of OHD implementation. OHD are the deliveries made outside of peak hours – that is, deliveries are made during late evening, night or early morning. Some of the interviewed BID representatives have provided assistance during the pre-study phase (such as participation in focus groups) and have actively supported the recruitment of businesses (goods receivers) that could participate in the OHD pilot project.

(5) Promotion of DSP among BID members. Designed to facilitate goods deliveries to specific destinations, the DSP is a managerial tool that drafts the measures that may positively impact the generation of externalities from deliveries while improving the quality of the service. DSPs can include consolidation of specific deliveries, reduction of trips, raising the load factors of delivery vehicles, eliminating carriers that are distant in relation to the end destination, collaboration

during procurement, and postulating the use of specific standards for delivery vehicles. One of the interviewed BIDs had conducted a pre-study for DSP implementation and a pilot project for several multi-tenant buildings located on one street in their area. Another BID is actively promoting DSP for the large office buildings in their area.

(6)   Several BIDS from the sample are promoting the use of collection points and locker banks for personal deliveries. The management teams of several BIDs expressed concerns about the extensive private deliveries made to the employees in the area, especially at big office buildings, which lead to situations wherein multiple delivery vehicles with one or few packages arrive at the building and are parked in front of it, sometimes for long periods of time because the delivery personnel have to find the package receiver. Some of the building managers are keen to ban personal deliveries to workplaces, and the use of collection points and locker banks could be a good alternative for the employees and for the particular area.

### 4.2. Property Owners

The commercial property owners were involved in the following activities that impact urban freight:

(1)   A big property owner, together with a logistics company, established a freight consolidation scheme and used electrical vans for the final deliveries from the UCC, which resulted in the significant reduction of deliveries received by the participating businesses.

(2)   Participation in a collaborative project about waste collection together with other stakeholders. This already-implemented project includes the property owner and a number of other stakeholders, such as a big service provider for recycling and waste management, and one carrier. This scheme combines waste collection with deliveries of packages, and the last mile delivery is operated by environmentally-friendly vehicles.

(3)   Consolidated delivery scheme initiated and implemented by a shopping mall owner. The scheme is implemented as follows: the goods for the participating retailers are delivered to the UCC outside the city centre and are delivered to the final customers by the collaborating logistics company using environmentally-friendly vehicles.

(4)   In a project in development, the property owner, together with other stakeholders (a waste management company, logistics company, and warehouse management company) took care of the waste and goods delivery for the designated area, including commercial and residential establishments, with the aim of decreasing transportation by up to 70% and using the freed-up space for greening, which is expected to increase the liveability and attractiveness of the area.

### 4.3. Facility Management Companies

The interviewed FMCs placed sustainability high in their operational agenda, and while they had activities in all four Nordic countries, their strategy was similar in each market. The following are the activities they were involved in and that impact urban freight:

(1)   Choosing suppliers with the lowest $CO_2$ transport impact. As part of the 'green' service provided to the customers, the FMC nominates the suppliers with the lowest environmental impact by considering, for example, the suppliers that deliver the products to the FMC's customers using greener vehicles or which have a more sustainable delivery strategy.

(2)   Tracking the cumulative effect of $CO_2$ emissions and reporting it to the customers on a yearly basis. The FMC that has been studied provides to the customers the cumulative report on the impact of total $CO_2$ emissions during the year, which includes the impact from the suppliers that delivered during this period; the impact of transport activities arranged by the FMC on behalf of the customers, including the transportation used by the FMC's site manager; and the reduction in $CO_2$ emissions achieved by the implementation of a continuous improvement program on-site.

(3)   Demand planning on behalf of the customers to minimise the delivery trips while keeping up the service level. The FMC site manager from the case study usually estimates the demand for each

product and product category and works out the delivery schedule together with the suppliers. The intention is to diminish the negative impact of deliveries while keeping the service level. For example, the replenishment strategy for stationery products at a big office building was set as once in two weeks.

(4) Implementation and management of common post rooms and concierge services. These services enable the receptionist or site manager to receive and send the post and packages on behalf of the customers (office tenants), resulting in decreased dwelling time for delivery vehicles and the consolidation of outbound flows of post and packages.

(5) Optimisation of the deliveries to foodservice establishments at their premises. One of the interviewed FMCs has worked out the canteen menu with the chef and optimised the delivery schedule for foods deliveries for the canteen in the multi-tenant office building, which resulted in a significant decrease in the number of delivery trips of foods without compromising the quality of food service provided to the tenants.

### 4.4. Public Procurement Organisations

The PPOs were involved in the following urban freight-related activities:

(1) Stipulating in the framework agreements that delivery vehicles must meet certain environmental standards. The supplier that delivers the products should ensure that the delivery vehicles meet the stipulated environmental standards (such as Euro 5 or Euro 6), regardless of whether the deliveries are made directly by the supplier or by a contracted carrier.

(2) Stipulating in the framework agreements the method and frequency of deliveries as well as penalising out-of-schedule and unplanned deliveries. For the transport-intensive deliveries, the purchaser and supplier agree on the delivery schedule; this puts the requirement on the final goods receiver (municipal establishment) to plan their demand in advance and communicate it to the supplier. Out-of-schedule deliveries can be arranged if needed, but the final customer is required to pay for additional unplanned transportation.

(3) Stating requirements for carriers to consolidate deliveries as much as possible. The aim is twofold: to minimise the dwelling time of delivery vehicles in the vicinity of institutions such as schools and kindergartens to decrease the potential for traffic accidents and to diminish the environmental impacts of deliveries. This was particularly relevant for transport-intensive deliveries, such as food deliveries to municipal institutions, and for carriers who have shown the capability to consolidate and deliver foods less frequently.

### 4.5. Qualitative Analysis

Our analysis of freight-related activities initiated or promoted by influencers identified the following as the most recurrent principal content:

(1) Vehicle-focused measures such as the usage of greener vehicles for the deliveries – that is, vehicles that have higher environmental standards, such as low-emissions vehicles, hybrid vehicles, and electric vehicles.

(2) Consolidation linked to physical infrastructure – that is, establishing a physical UCC, and in connection with the UCC, setting up delivery solutions for the last mile.

(3) Consolidation through behavioural changes, including activities that imply the implementation and use of collaborative procurement schemes, demand planning activities and goods consolidation that should lead to the consolidation of deliveries.

The analysis focuses on these three types of activities and their potential influence on the business models of carriers.

### 4.5.1. Vehicle-Focused Measures

Each of the influencing organisations promoted or initiated the implementation of deliveries services utilising greener vehicles by either directly stating standards for vehicles or supporting the choice of carriers with specific types of vehicles, such as electric vehicles. When these measures are invoked, the carriers have no choice but to comply or lose the contract. Hence, carriers will be forced to make operational changes that may, depending on the vehicle or vehicle solution, impact how their business model functions.

The analysis of the BMC showed that each block, except the key activities and revenue streams, was directly influenced by initiatives linked to greener vehicles. This does not mean that the key activities will not be influenced. For example, a switch to an electric vehicle may change parking-related activities by enabling parking in specific areas designed for the vehicle. However, changes associated with the vehicle will likely not impact the key activities associated with the direct interaction between the carrier and the customer or the organisational aspects of the deliveries. For the carrier, changes to the vehicle or the adoption of a new vehicle type will possibly be accompanied by changes in its relationship with the vehicle supplier and maintenance services. Changes to the vehicle pool add to or change the key resources that the carrier relies on. It may also mean that drivers and service staff might need specific training. The significance of such changes is difficult to assess. However, deliveries with greener vehicles do add to the value proposition that both the shipper and receiver may benefit from.

The customers' segmentation, customers' relations, and channels will be affected as well, potentially pressuring the carriers to seek long-term relations with the customers to lower the risks associated with investments into new delivery vehicles. In the case of an electric vehicle or an additional vehicle designed for or designated to a specific service area, the cost structure shifts to fixed costs owing to the rising capital expenditure. Revenue streams are not affected unless influencers or receivers decide to impose some type of incentives or disincentives linked to the vehicle category.

### 4.5.2. Consolidation Linked to Physical Infrastructure

The method via which the promotion and implementation of the physical consolidation infrastructure, such as a UCC or common post rooms, is achieved differs among the categories of influencers and varies with their ability to make direct changes or to indirectly influence changes to the physical environment within their area of interest. Consequently, influencers with a greater affinity to, and control over, the physical infrastructure may be more eager to promote and facilitate initiatives that support consolidation through physical investments. Changes to the physical infrastructure are comparatively more difficult to bypass than the consolidation through behavioural changes. The environmental effect of such initiatives can lead to decreased total travel distances for the goods delivered, an increased load factor of freight vehicles, diminished traffic congestion, and free space on highly-congested streets. Such schemes usually involve the last mile delivery using greener vehicles, which altogether positively impacts the sustainability of urban freight.

When examining the potential influences of these changes, the canvas showed that each block in the business models of carriers is potentially affected by the implementation of the UCC scheme, such as changing the key partners (in the case of adding new partners linked to the management of a UCC), changing the key activities and value proposition (as the goods are going to be re-routed to a UCC, which means that the carrier loses the interaction with the end customer), changes in the key resources (freeing up time because deliveries to fixed locations such as postal lockers or UCCs are quicker and easier to plan than deliveries to end customers), changes in customer segmentation and customer relations (no direct contact with end customers and a different type of channel), and cost structure (reduced costs associated with the last mile) and revenue streams (because of the necessity to share revenues with the final delivery carrier). The implementation of physical consolidation initiatives may lead to significant changes in the business models of carriers and may, especially considering the loss of customer interaction, lead to friction between influencers and carriers.

### 4.5.3. Consolidation Through Behavioural Changes

Different influencers promote behavioural changes differently and with varying levels of commitment. For example, PPOs actively interject changes in the procurement process and strive for demand planning, demand consolidation and consolidating deliveries. BIDs and FMCs focus on sustainable procurement and bundle procurement. However, in this study, property owners did not interfere with the procurement and consolidation processes; thus, there was no potential for property owners to impact consolidation behaviour in this way. Consolidation of flows through behavioural changes can lead to less freight traffic coming in, going out, and within the urban area. This can lead to lower traffic congestion, freed loading and unloading zones, more parking spots, and the diminishing of other externalities caused by urban freight. The implementation of consolidated deliveries and collaborative procurement schemes has several potential effects on the business models of carriers. Because consolidation through behavioural changes may occur in different stages of the procurement or delivery process, it is difficult to identify if there will be changes to the key partners that the carriers rely on; however, our data suggest that this would not be the case because the focus lies on the coordination of actors among receivers or shippers rather than on the carrier stage in the supply chain.

Consolidation and coordination may change the key activities and resources as well as involve additional handling of goods. The value proposition changes because consolidation ensures scheduled deliveries instead of frequent and comparatively less-planned stops. Customer segmentation and customer relations may change because this type of consolidation divides the customer base according to delivery preferences and will disrupt the already established coordination and communication patterns. Revenues may be influenced through changes in fee structures and the decreased number of opportunities to sell additional services. The costs structure may change; the increased fill rates and decreased number of unplanned deliveries will increase the savings of carriers. However, depending on who handles the packages' additional handling, the costs for carriers may increase.

### 4.6. Potential for Financial Impact on the Business Models of Carriers

The initiatives have the potential to negatively impact the financial viability of the carriers' business models. The introduction of a greener vehicle leads to investments that increase the capital costs of the carrier and, in the case that the carrier owns the vehicle, increases the fixed costs, which means that the operational risk for the carrier increases comparatively more than that in the case of organisational initiatives. In the case of initiatives of an organisational nature, there are costs associated with learning and staffing; although such costs may be considerable, they are often easier to counteract (e.g., externalise, postpone or gather support among other stakeholders) and tend to be of the variable type, meaning that they are easier to manage in the case of a fall in customer demand. Consequently, the organisational initiatives do not increase the operational risk in the same way as the initiatives that imply significant physical changes.

The second type of activity, the establishment and use of a physical infrastructure for the consolidation of goods, aims to decrease traffic pressure in certain areas. The establishment of a UCC, where the goods destined to certain areas are accumulated and sent to the receivers through a single flow, can be costly and challenging [38]. The practice of establishing a UCC and other physical infrastructure such as lockers adds both fixed and variable costs (e.g., rents, handling, loading and unloading). Such solutions are complicated to introduce because they drastically change the business models of many well-established carriers. First, the operator that consolidates the flows and makes the final deliveries must find a warehouse (usually not far away from the city) and then organise the flows (both deliveries and pick-ups) often several times a day. Furthermore, shippers and carriers must give up the last mile and leave goods at the UCC. Carriers may thus lose opportunities for face-to-face interaction with their customers, which may change the customer relationship and potentially decrease future revenues.

The third type of initiative entails changes to carriers schedules which, in the case of deliveries outside of normal work hours, means re-arranging staffing and, in the case of night-time deliveries, means implementing specific equipment (e.g., quieter cargo cages or vehicles), resulting in increased

costs [39]. In countries such as Sweden with strong unions or strict legislation, changes in arrangements may result in considerably higher costs associated with wages and compensations. In addition, there must be a way for the receiver to accept and verify the goods. Having a person at the receiver's site that accepts the goods is costly; however, arranging for the carrier to have access to the building requires a trustful relationship between receivers and transporters as well as technological systems such as CCTVs and electrical lockers. Because the value proposition that carriers provide to their customers is focused on timeliness and reliability, carriers may need to establish even closer relations with the goods receivers, such as a common IT system that can help them synchronise the delivery and goods acceptance.

A cumulative summary of the changes in the business models of carriers from our cases is given in a canvas (see Table 2).

**Table 2.** The cumulative changes in business models of carriers.

| Building blocks | Changes in business model |
|---|---|
| *Infrastructure* | |
| Key partners | Changes in key partners: adding influencing organisations such as BIDs or property owners as a key partner, adding a consolidation centre coordinator in the case of consolidated deliveries or acquiring a new vehicle supplier or new maintenance service provider. |
| Key activities | Changes in key activities: adding consolidation activities (with additional handling or/and coordination) for specific groups of goods, specific goods receivers, specific areas, etc. In case of OHD, operating during the night, including the difference in operation for unattended deliveries. Deliveries to UCC also bring changes in the key activities. |
| Key resources | Key resources such as vehicles or the physical location may change to comply with the environmental standards; ICT software may be needed for scheduling the deliveries and consolidating the goods. More resources dedicated to planning and coordination of the flows. |
| *Offering* | |
| Value proposition | Changes in value proposition for the customers, for example, in the case of scheduled deliveries, consolidated deliveries and providing more sustainable deliveries to the customers using greener vehicles. The carriers delivering only to the UCC and not to the final customer also change their value proposition. |
| *Customers* | |
| Customer relationships and channels | Entering into more long-term relations with customers in case of special agreements with them on providing extra service such as deliveries by environmentally friendly vehicles, consolidated deliveries and deliveries from the UCC to final goods receivers. Expanding the coordination and communication with goods receivers and shippers or in some cases with influencing organisations. |
| Customer segments | The customer segments may change. The customers are going to be segmented according to their requirements for deliveries and consolidation, being coordinated by influencing organisations, adding a new intermediate or coordination instance as influencing organisations for example. |
| *Finances* | |
| Cost structure | The cost structure will change owing to high investments into 'greener' vehicles, the necessity of sharing the income flows with other carriers, delivering for the final mile or, in case of consolidated deliveries, with UCC, etc. Possibility to decrease running costs per package in case of better vehicle filling rate. Additional handling for consolidation might increase costs. |
| Revenue streams | Revenue streams change owing to changes in, and implementation of, differentiated delivery schemes. The use of a UCC with a different carrier for different stages impacts how each carrier gets compensated and the ability of carriers to change fees or contracts. Negotiation between the carrier and the receiver may change by the influencer injecting itself, or specific rules, in the negotiation process. |

In summary, the influencing organisations are different, and they influence the business model of carriers in different ways. The analysis shows that the type of activity or change results in a higher impact on business models of carriers than the type of influencer that initiates it. However, in our cases, different types of influencers tend to engage in certain types of activities. Our empirical data show that BIDs, FMCs, and PPOs exert their influence by promoting demand planning, consolidated deliveries and collaborative procurement (also sustainable procurement). The property owners are focused on reorganising and re-arranging the deliveries using UCC or similar establishments. Understanding the changes in the business models of carriers produced by certain freight activities can help to understand the barriers of the changes and how they can be reduced.

## 5. Discussion

The nature of the influence exerted by influencers is decided by their modus operandi as well as the type of goals that they have regarding urban freight issues. Influencers have different reach in relation to the goods receivers; some influencers set the rules that carriers must obey, others suggest how to organise flows differently, or act as an orchestrator that consolidate the flows [35,36]. Thus, the potential for influence on the business models of carriers differs considerably depending on the agenda and ambition that the influencer sets for itself. Interestingly, the incorporation of sustainability as a key aspect of logistics services leads to an expansion of this agenda. Consequently, as the pressure rises on influencers to incorporate sustainability into their own goals, the potential for these organisations to impact the business models of carriers should be expected to rise. Influencers may thus be categorised in relation to the range of services they offer and the geographical area in which they are active: that is, whether they provide a narrow or wide range of services to a geographically-clustered or dispersed group of clients. These categorisations imply that the impact on the business models of carriers may increase both in likelihood and severity when discussing influencers that supply a wide range of services to actors that are geographically-dispersed.

An aspect to consider when analysing the work of influencers is the role that they play in relation to the future trajectory of the urban freight sector. Influencers may have more political clout and access to a larger and more diverse resource base than both receivers and carriers [2]. Acting on behalf of the groups that they represent, influencers may thus voice concerns on a wide range of forums. Some of the influencers are also involved in the planning of the built environment; this means that they are active during more phases associated with the development and management of the urban community. This implies that influencers may support sustainability in ways that are impossible for both receivers and carriers. Influencers can also actively pursue and participate in projects and programs for sustainable development. This means that they impact the early-stage development of sustainable innovations, giving them more weight than what is expected when considering their lack of direct presence in the supply chain.

Likewise, because influencers are formulating and initiating requirements, sometimes coupled with penalties, associated with deliveries and their function as coordinators for such requests within a geographical area, it is possible that carriers may come to see influencers as a new type of key partner. Furthermore, because the influencers' interests are associated with specific geographical areas and the aspects linked to their performance, it is possible that they have a more long-term perspective on issues associated with freight deliveries, thus making them more motivated to have a clear goal to which carriers may align their strategic vision.

The results also indicate that hesitance towards sustainability efforts among carriers may be better understood by evaluating how a specific action will impact the existing business models. By doing so, it is possible to identify the potential bottlenecks associated with the implementation of a project or other urban freight initiatives. Such knowledge may help to establish a constructive dialogue between the central stakeholders (receivers, shippers, influencers and carriers) to drive the changes for more sustainable urban freight. Another application could be to rank activities according to their impact on the business model and to choose to focus on those that correspond to the available resources. Carriers cannot be expected to bear the entire cost of the implementation of innovative and more sustainable solutions for urban logistics. A dialogue may be the way to reach an agreement, share the responsibility among the stakeholders and help to overcome the barriers by collective agreement and actions, such as sharing additional costs.

Even if one should be cautious to make generalisations based on a small and heterogeneous sample, this study presents findings which may indicate promising pathways for future research. While the agenda and involvement in urban freight questions may differ, the organisations studied are present in most dominant markets around the world. However, the results suggest that stakeholders and actors in cities and countries with different cultural and political setup have contradicting interests [20] but similar perceptions about transportation and urban freight [40]. Furthermore, the level of awareness about sustainability issues, together with the power that influencers have over other stakeholders and actors, can be incentivised and supported by policy-makers that want to foster a sustainable urban freight sector.

## 6. Conclusions

This study focuses on a category of indirect stakeholders, called influencers, who should have considerable responsibilities associated with the space in which deliveries are made and who also shape the norms and regulations which moderate the behaviour of commercial actors in the urban freight setting. The role that influencers play in the urban freight setting has, however, been an understudied topic. Hence, the purpose was to explore what roles that influencers play in the development of sustainable urban freight services and to identify how the different initiatives that those stakeholders launch may impact the business models of carriers.

Based on a review of research on the interactions between the stakeholders within the urban freight sector, the classification of such organisations and their potential for influencing business models of carriers were compiled. The study shows the variety of these types of organisations as well as the differences in their modus operandi, that is, the way in which they interact with and influence other organisations. However, one common feature is that influencers can reach and bring together many fragmented goods receivers, thus promoting and encouraging sustainable logistics while providing a platform to organise sustainable solutions.

The following are the main findings of the study:

(i)     The influencers play an important role in promoting sustainable urban freight activities because they influence different groups of goods receivers.

(ii)    The influencers have different relations with the goods receivers and their impacts on delivery practices differ.

(iii)   Different types of influencers tend to engage in different types of sustainable urban freight activities based on the influencer's own resources, goals and range of activities.

(iv)    The business models of carriers are likely to be impacted by the activities promoted or initiated by influencers, and the impact is greater when influencers have power through stronger presence or more comprehensive rules.

(v)     The potential impact on the business models of carriers depends more on the type of activity and less on the type of influencer that initiates the sustainable urban freight activity; however, the potential for conflicts with carriers arises owing to the probable costs, including investments and operational changes, that business model changes entail.

In sum, the research highlights how influencers may directly or indirectly pressure receivers to raise the environmental performance of freight solutions. In doing so, influencers impact the behaviour of other actors in the urban freight setting and facilitate the development of more sustainable and attractive cities.

Future research may expand on the classification of influencing organisations, list their activities, and add recommendations to the stakeholders and/or policymakers that are interested in fostering sustainable urban freight systems. Furthermore, a possible next step is to assess, through surveys or case studies, how the activities identified in this study impact the carriers that provide transport services in areas where influencers have a mandate to impact delivery conditions. By taking these steps it would be possible to develop a richer understanding of how influencers may facilitate sustainable delivery practices.

**Author Contributions:** Conceptualization, A.B. and J.W.; methodology, A.B.; formal analysis, A.B.; investigation, A.B. and J.W.; data curation, A.B.; writing—original draft preparation, A.B. and J.W.; writing—review and editing, A.B. and J.W. All authors have read and agreed to the published version of the manuscript.

**Funding:** This research received no extra funding.

**Acknowledgments:** We are grateful to all interviewees that participated in the study for their openness and willingness to share their knowledge. We also would like to thank those that supported the research process by giving feedback and helping us getting in contact with respondents.

**Conflicts of Interest:** The authors declare no conflict of interest.

# Appendix A

| Sections | Quotes |
|---|---|
| | **4.1. Business Improvement Districts** |
| (1) | "We are, like all other BIDs, looking into how we consolidate and how we reduce the impact of deliveries. When you look at deliveries of stationary alone it is something we have picked up on. I mean businesses like these could have three or four stationary suppliers bringing them things every single day since one department is unaware what another department is ordering. So, what we have got in places is an agreement with the stationary company. Because we talked to them as a group, and encouraged our members to go to them, to receive a discounted service. That is why there we could save although it is not a consolidation, but we are reducing the impact". <br> BID 1 in UK |
| (2) | "So, we have been running the [street name] project for the number of years, almost three years now, and it was an initially the pilot on waste consolidation and then we added goods deliveries to it". <br> BID 3 in UK |
| (3) | "But there is an opportunity to use that UCC and then to talk to carriers with electrical vehicles here to have goods delivered and picked up in this area". <br> BID 1 in UK |
| (4) | (About off-hour deliveries) " … I always recommend having [deliveries] done in the middle of the night. And what I'll do is, I'll work with [the persons involved] to make sure that the drivers don't get a ticket because, that's what happens. On the road everyone is parking illegally to do this, to do these deliveries in the middle of the night. If you could just relax enforcement, then it goes a long way, because if every time you get a delivery, you also get a $100 ticket, that will be $5000 a week". <br> BID 1 in US |
| (5) | "So, we have done several DSPs, delivery service planning, on physical street areas, like [street name], where we worked with about 30 of the local shops and businesses". <br> BID 2 in UK |
| (6) | "We try to do as much as possible with the occupants around click and collect, trying to encourage personal deliveries to not to come in to [the city centre]". <br> BID 2 in UK |
| | **4.2 Property owners** |
| (1) | "One of the examples is freight consolidation scheme, introduced at one of the most important streets here; it is one of the successful projects initiated and supported by this property owner together with other stakeholders". <br> BID 3 UK about property owner, UK |
| (2) | "We have been involved in the project in Stockholm with the entrepreneur I have been talking about, in our head offices, we have a service actually, it is called "love your city", it is located here, this is one big property, so goods are coming here and the entrepreneur, they distribute the goods to others in this area". <br> Property owner, Sweden |
| (3) | "We want to contribute to the city accessibility and sustainability". <br> Property owner, Sweden |
| (4) | "We think that to be at one place is very good because then we have a lot of influence on the neighbourhoods, which we actually like, and we think we are good at it. And in that specific case we figured out if we could handle the last mile, we could probably decrease the transportation by up to 70% or something. [ … ] we made a survey from which we could figure out that if we don't do anything the yearly number of transports in this area will be around 400 000. If we could cut that down to 100 000, it would be a much better area to live in, or to work in, or to visit. We could then use the streets and green areas for other things. [ … ] we could raise the utilization ratio while still being able to keep the green areas [ … ] So with that idea in mind we started this company". <br> Property owner 4, Sweden |
| | **4.3 Facility Management Companies** |
| (1) | "Our suppliers that we use are quite strictly controlled. Not any supplier can become our sub-supplier. For example, a coffee supplier designs their route when they drive to the city, and it should be as effective as possible. Then they even report how much CO2 emissions they produce. They report that to me [ … ] and we then report that in our system". <br> FMC site 1 |
| (2) | " I report all my work trips with my work car, once a year. How much I was traveling for work and how much emission was produced by my car. Because we have a responsibility to report in order to keep our certificates". <br> FMC site 1 |
| (3) | "I, and the people that work on the site, put together all requests into one order. For example, with stationary, we conduct inventory and then add to the list what should be refilled, and then we order from the list once in a month". <br> FMC site 1 |
| (4) | "And then we have of course the thing called "[company name] smart flow". It is for postal services within a building. So, instead of the post officers delivering the post to the building, they give all the packages to us and we sort them and send email to the customers to come down and pick up". <br> FMC site 2 |
| (5) | " We consolidate the deliveries as well. So instead of like the regular restaurants that have around 15 providers, we have 4. And we only take deliveries like, the big deliveries, only twice a week. And then we have like fish and stuff like that – that is daily. We actually agreed to pay the provider an environmental fee, since that makes my head chefs more eager to consolidate the transports". <br> FMC site 2 |
| | **4.4 Public Procurement Organisations** |
| (1) | "Yes, well when it comes to transport-intensive deliveries, we can put requirements on the vehicles. For example, the trucks used for distribution must be Euro 5 or other types of requirements. If they drive in the centre, they have to use biofuels and nothing else". <br> PPO 1, Sweden |
| (2) | " … then there were requirements on how the deliveries are done. For example, what type of vehicle that is used, how often the deliveries are done. We can put requirements both when it comes to deliveries which are free of charge or paid for by the goods receivers, so those are the things that we can make requirements about. This means that we do not have any fast or unplanned delivers". <br> PPO 1, Sweden |
| (3) | "We do not want to have a large number of delivery trucks that drive around in the vicinity of the school yard, it is a security issue. So, we try to control this fairly meticulously". <br> PPO 1, Sweden |

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
