# Peer review of "The Role of ‘Influencers’ as Drivers of a More Sustainable Urban Freight Sector"

_sustainability, doi:10.3390/su12072850_

Round 1

Reviewer 1 Report

I fount the paper very interesting and innovative, written using academic standards. The authors contain all necesary aspects of the problem.

In my opinion paper can be published as it is.

Author Response

Thank you for reading our paper.

Reviewer 2 Report

I would like to praise the authors for the good paper submitted which addresses a relevant aspect in urban freight distribution. Some general considerations and some specific ones.

general --> (a) the considerations reported are described as if they are relevant for all the sectors and the circumstances. This can/should be improved considering the heterogeneity in the (relatively) small sample considered. It would be nice to have additional characterization of the considerations, descriptions an, most importantly, policies/suggestions/implications described. (b) most of the considerations reported as described as self evident, This can/should be improved. In particular this can be achieved citing (e.g. verbatim) some sources discussing/suggesting a specific issue. This should be duly taken into consideration when discussing results and promoting possible intervention policies

specific --> there are some relevant contributions in the liteature, especially with respect to the behavioural change approach that have not been citied. They should so to provide the reader with a more comprehensive picture of the approach/resuts obtained. In fact, while BMC is a perfectly fine instrument to investigate the interactions and the heterogeneity in preferences among the relevant stakeholders in the urtban freight distribution system, one should also recognise that this approach is, in nature, qualitative and can only provide some rough descripotions/suggestions at a strategic level hwreas it lacks the capability to provide decison makers with more quantitati, down-to-earth detailed implications wrt to the policies they might want to define and deploy.

In particular I suggest including the following articles:

  • Role-playing games as a mean to validate agent-based models: An application to stakeholder-driven urban freight transport policy-making, M Le Pira, E Marcucci, V Gatta, Transportation Research Procedia 27, 404-411
  • Urban freight transport and policy changes: Improving decision makers' awareness via an agent-specific approach, V Gatta, E Marcucci, Transport policy 36, 248-252
  • Simulating participatory urban freight transport policy-making: Accounting for heterogeneous stakeholders’ preferences and interaction effects, E Marcucci, M Le Pira, V Gatta, G Inturri, M Ignaccolo, A Pluchino, Transportation Research Part E: Logistics and ransportation Review 103, 69-86
  • Towards a decision-support procedure to foster stakeholder involvement and acceptability of urban freight transport policies, M Le Pira, E Marcucci, V Gatta, M Ignaccolo, G Inturri, A Pluchino, European Transport Research Review 9 (4), 54.

Author Response

Reviewer 2

Comments

How we have addressed the comments

the considerations reported are described as if they are relevant for all the sectors and the circumstances. This can/should be improved considering the heterogeneity in the (relatively) small sample considered.

It would be nice to have additional characterization of the considerations, descriptions an, most importantly, policies/suggestions/implications described.

Thank you very much for the comment. Indeed, some additional explanations on research relevance are needed. We have added some additional explanations on the difference between certain influencers in different countries (lines 151-156). Additional explanations on the choice of organisations for empirical data collection are added (lines 248-253).

We have also added some reflections on the relevance of the results and considerations for policy suggestions (lines 588-596). 

most of the considerations reported as described as self evident, This can/should be improved. In particular this can be achieved citing (e.g. verbatim) some sources discussing/suggesting a specific issue. This should be duly taken into consideration when discussing results and promoting possible intervention policies.

Thank you very much for your comment. In order to support the summarized results we have created a table with the citations corresponding to the results. We suggest to place this table at the appendix (see Appendix 1) but it can be moved to the results part if suggested by the reviewers and editors.

specific --> there are some relevant contributions in the literature, especially with respect to the behavioural change approach that have not been citied. They should so to provide the reader with a more comprehensive picture of the approach/resuts obtained. In fact, while BMC is a perfectly fine instrument to investigate the interactions and the heterogeneity in preferences among the relevant stakeholders in the urtban freight distribution system, one should also recognise that this approach is, in nature, qualitative and can only provide some rough descripotions/suggestions at a strategic level hwreas it lacks the capability to provide decison makers with more quantitati, down-to-earth detailed implications wrt to the policies they might want to define and deploy

In particular I suggest including the following articles:

·         Role-playing games as a mean to validate agent-based models: An application to stakeholder-driven urban freight transport policy-making, M Le Pira, E Marcucci, V Gatta, Transportation Research Procedia 27, 404-411

·         Urban freight transport and policy changes: Improving decision makers' awareness via an agent-specific approach, V Gatta, E Marcucci, Transport policy 36, 248-252

·         Simulating participatory urban freight transport policy-making: Accounting for heterogeneous stakeholders’ preferences and interaction effects, E Marcucci, M Le Pira, V Gatta, G Inturri, M Ignaccolo, A Pluchino, Transportation Research Part E: Logistics and ransportation Review 103, 69-86

·         Towards a decision-support procedure to foster stakeholder involvement and acceptability of urban freight transport policies, M Le Pira, E Marcucci, V Gatta, M Ignaccolo, G Inturri, A Pluchino, European Transport Research Review 9 (4), 54.

Thank you very much for your comment. We agree that adding relevant literature provides reader with better understanding of the topic and improves the quality of the paper in general. Thank you for the list of literature sources proposed, that was really helpful. We have added some of the literature sources proposed to the literature section (lines 111-113).

Reviewer 3 Report

The paper is very interesting and presents very important and in some way novel topic regarding role of influencers on sustainable urban freight delivery. So far, many papers focus on the role of various stakeholders but rarely are focusing on this specific group which was selected by the authors.

The title is adequate to the content of the paper and presents the importance of influencers on sustainable urban freight sector.

In the abstract the authors formulate the statement “third-party urban freight stakeholders”. In logistics dictionary the word third-party is related mainly to third-party logistics service providers. I think that this statement without explanation can confuse a reader. I understand the authors’ intention. However, I think it should be a short explanation of this meaning already in the abstract.

The purpose introduced in the abstract should be more coherent to the purpose presented in the introduction. In the introduction you mention business models of the carriers involved in UFT and while the abstract you didn’t . It seems to be important while in literature review you focus on business models.

Section 2. (literature review). In my opinion you shouldn’t separate stakeholders and influencers because they form the same group. Regarding the title of the paper you should more focus in this part of the paper on influencers – including this specific group selected for the research (as a separated group of stakeholders)

I think that the research procedure should be more thorough explain. You mentioned that you interviewed 20 organizations including 15 influencing organizations. So what kind of organizations are those 5 (not influencing)? It is hard to follow the description of the interviewed respondents. Regarding description of the analyses in my opinion the text could more clear if all steps were presented as a bullet list.

I propose to change the title of the section 4.5 Analysis to 4.5. Qualitative analyses

In the section 5. Discussion - there is a lack of comparison to other research presented in the literature

In the section 8.Conclusions – you wrote “Based on a review of previous research on the interactions between the stakeholders within the urban freight sector, the classification of such organisations and their potential for influencing business models of carriers were compiled”. – what do you mean previous research? Previous to those you introduced in the research? I think that you should avoid word “previous” .

Minor remarks:

Please do not use shortcuts in titles of sections

Table 1. (two times “word” carriers in “Business model of carriers and carriers)

Lines: 527-530: “This study aimed to explore the roles that third-party stakeholders, who have considerable responsibilities linked to the norms regulating local delivery areas and are thus referred to as influencers with regard to other actors in the urban freight setting, play in the sustainability of urban freight services and the business models of carriers” – please modify this sentence because is too long and not clear enough.

Author Response

Comments

How we have addressed the comments

In the abstract the authors formulate the statement “third-party urban freight stakeholders”. In logistics dictionary the word third-party is related mainly to third-party logistics service providers. I think that this statement without explanation can confuse a reader. I understand the authors’ intention. However, I think it should be a short explanation of this meaning already in the abstract.

Thank you for your comment. Indeed, our formulation “third-party urban freight stakeholders” might create confusion for the readers. We changed it to formulation as “stakeholders with indirect interests in urban freight” (line 14).

The purpose introduced in the abstract should be more coherent to the purpose presented in the introduction. In the introduction you mention business models of the carriers involved in UFT and while the abstract you didn’t . It seems to be important while in literature review you focus on business models.

Thank you for the comment. Indeed, it is important to mention business models of carriers already in the abstract. We have corrected that (line 23).

Section 2. (literature review). In my opinion you shouldn’t separate stakeholders and influencers because they form the same group.

Thank you for the comment. We agree that we should make it more clear, that we do not separate stakeholders and influencers, because influencers are a specific group of stakeholders. However, we would like to make a clear focus on this particular type of stakeholders in this research as we believe that they play an important role and they are often overlooked by researchers and policy-makers. We added this clarification at lines 122-124.

Regarding the title of the paper you should more focus in this part of the paper on influencers – including this specific group selected for the research (as a separated group of stakeholders)

We agree on this comment about the focus on influencing organisations. We changed the title of the section to “Research on Influencing organisations as a group of stakeholders in urban freight”. We think the new title is more coherent with the content of the section and the focus of the paper.  Line 104.

I think that the research procedure should be more thorough explain. You mentioned that you interviewed 20 organizations including 15 influencing organizations. So what kind of organizations are those 5 (not influencing)? It is hard to follow the description of the interviewed respondents.                                             

                                                                           Regarding description of the analyses in my opinion the text could more clear if all steps were presented as a bullet list.

Thank you for the comment. We agree that it was not very clear. We have inserted the clarification about influencers and the rest of interviewees, lines 231-234. In order to make it easier to follow the description of interviewed respondents, we created the table that lists the organisations that we interviewed for the study, Table 1, line 236.

Thank you for the comment, we agree with the suggestion and we created the diagram (Figure 2) that shows how the research process was organised (line 257).

I propose to change the title of the section 4.5 Analysis to 4.5. Qualitative analyses

Thank you for the comment, we have changed the title of this section to “Qualitative analysis” (line 404).

In the section 5. Discussion - there is a lack of comparison to other research presented in the literature

Thank you for your comment, we agree on it and have added some sources from the literature section to the discussion section. The paper looks more academically coherent now (lines 547-8, 561, 593-4).

In the section 8.Conclusions – you wrote “Based on a review of previous research on the interactions between the stakeholders within the urban freight sector, the classification of such organisations and their potential for influencing business models of carriers were compiled”. – what do you mean previous research? Previous to those you introduced in the research? I think that you should avoid word “previous” .

Thank you for your comment, we agree that using the word “previous” creates confusion for the reader. We have deleted this word (line 606).

Please do not use shortcuts in titles of sections

We agree on this comment, we have corrected that (lines 291, 352, 383).

Table 1. (two times “word” carriers in “Business model of carriers and carriers)

Thank you for careful reading, we have corrected that now (line 532).

Lines: 527-530: “This study aimed to explore the roles that third-party stakeholders, who have considerable responsibilities linked to the norms regulating local delivery areas and are thus referred to as influencers with regard to other actors in the urban freight setting, play in the sustainability of urban freight services and the business models of carriers” – please modify this sentence because is too long and not clear enough

Thank you for the comment, we have modified this sentence, we hope it is more clear now (lines 599-603).

Round 2

Reviewer 2 Report

I am satisfied with the changes brought to the paper